# Analysis of and Experimental Research on a Hydraulic Traction System Based on a Digital Hydraulic Transformer

**DOI:** 10.3390/s22103624

**Published:** 2022-05-10

**Authors:** Weijian Li, Zhuxin Zhang, Tao Liu, Hengyi Cao, Tao Ni, Yafei Wang

**Affiliations:** 1School of Mechanical Engineering, Yanshan University, Qinhuangdao 066004, China; lwj-mh@stumail.ysu.edu.cn (W.L.); liutao@ysu.edu.cn (T.L.); chy0608@stumail.ysu.edu.cn (H.C.); 2Hebei Key Laboratory of Special Delivery Equipment, Yanshan University, Qinhuangdao 066004, China; nitao@ysu.edu.cn; 3School of Vehicle and Energy, Yanshan University, Qinhuangdao 066004, China; 4CSSC Systems Engineering Research Institute, Beijing 100094, China; yfwang_csu@163.com

**Keywords:** hydraulic transformer, digital control, traction, flow control, secondary regulation

## Abstract

In this study, we designed a new type of digital hydraulic transformer using four gear–pump/motor units with a displacement ratio of 2^0^:2^1^:2^2^:2^3^ and two control valve groups that consist of four solenoid directional valves. The driving gear shafts of the four gear–pump/motor units are fixedly connected to achieve synchronous rotation. The two control valve groups are respectively installed through an integrated valve block on the inlet and outlet of each gear–pump/motor unit. With the objective of reducing the installed power and energy consumption of hydraulic traction systems, we propose a new energy-saving hydraulic system based on a digital hydraulic transformer. This hydraulic system uses a digital hydraulic transformer as a pressure/flow control element. By controlling the power on/off states of eight solenoid directional valves, the digital hydraulic transformer can realize a change in output flow and then a change in speed of the hydraulic cylinder piston rod. Through the theoretical derivation and simulation analysis of the hydraulic system pressure/flow change process, and the experimental verification of the built hydraulic traction system based on the experimental platform, a conclusion is drawn that the proposed digital hydraulic transformer can change the output pressure/flow of a hydraulic system through a binary digital control, verifying the feasibility of the pressure change principle of the designed digital hydraulic transformer and the rationality of the hydraulic traction system circuit.

## 1. Introduction

A hydraulic transformer is a hydraulic secondary component that can convert hydraulic energy into mechanical energy and vice versa. In hydraulic transmission, the output pressure of the common pressure rail (CPR) system can be theoretically adjusted to the pressure required by the load without throttling loss [1,2,3]. Existing hydraulic transformers are classified into linear and rotary types [4,5]. The simplest structure of a linear hydraulic transformer is the booster cylinder, in which the piston rods of two single-rod hydraulic cylinders are rigidly connected; the pressure transformation ratio of the booster cylinder can only be a fixed value due to the structural design of this hydraulic transformer [6]. Li et al. proposed a digital hydraulic transformer whose pressure transformation ratio can be changed discretely in accordance with a pattern of natural numbers through combinations of the piston area in multistage hydraulic cylinders; the utilization ratio of hydraulic cylinders and the output fluctuation of hydraulic control units were analyzed [7]. However, this hydraulic transformer cannot be controlled to achieve continuous pressure output and can only be applied to cases with a small flow due to the limitation of its structural design. Rotary hydraulic transformers are currently classified into plunger and vane types. Kouns proposed a plunger hydraulic transformer with bidirectional pressure variation in which two independent axial plunger pumps/motors are mechanically connected by their rotors; this transformer can change output pressure by adjusting the displacement of the plunger body. This hydraulic component is known as the traditional hydraulic transformer. However, this type of hydraulic transformer exhibits the disadvantages of large size, complex structure, large leakage and friction, and low efficiency; consequently, such transformers are seldom used at present. A new type of hydraulic transformer was studied by Achten et al. up to its third generation [8,9]. This type of hydraulic transformer combines the functions of a plunger pump and a plunger motor into a composite design. The evolution process is primarily from a change in the number of plungers and the structure of the cylinder to the optimization of the flow distribution plate structure to reduce noise and improve work efficiency. Liu et al. designed an electrohydraulic serve swash plate axial plunger-type transformer to change the pressure transformation ratio of a hydraulic transformer by controlling the oscillating motor [10,11,12]. To solve the problems of hydraulic transformers, including high fluctuation of output flow and pressure, inability to operate at low speed, and high noise, Liu et al. proposed a novel configuration of hydraulic transformers with double rotors and discussed the influences of structural parameters (e.g., the wrap angle of the distribution slot and the number of plungers) and operating parameters (e.g., control angle and CPR pressure level) on the characteristics of a hydraulic transformer [13,14]. Yang et al. proposed a variable hydraulic transformer for realizing the simultaneous control of load pressure and load flow of a new hydraulic transformer, effectively reducing noise and leakage and further improving the pressure transformation ratio [15,16,17]. Another problem is that the inlet and outlet also change when an existing new hydraulic transformer realizes the function of pressure transformation. To address this problem, Zhou et al. proposed an oblique axis piston four-port hydraulic transformer [18,19,20] and tested the pressure transformation ratio. Zang designed a vane-type hydraulic transformer in 2016 and theoretically analyzed it on the basis of a non-common pressure rail secondary regulation system [21]. The current research on a new hydraulic transformer is primarily focused on its control strategy, noise reduction, and improving its range of pressure transformation ratio.

In principle, the structural design of the aforementioned hydraulic transformers includes three oil ports: the oil source, load oil, and low-pressure oil ports. The principle of pressure transformation involves absorbing/releasing part of the low-pressure oil port flow to increase/reduce the flow of the output oil port, and consequently, reduce/increase output pressure. Liu et al. proposed digital pumps/motors, i.e., a multi-gear pump stepped variable system [22]. Digital pumps/motors are composed of three or more quantitative pumps/motors with the same or different displacements. The output shafts of all quantitative pumps/motors are fixedly connected. When the input rotational speed and output displacement of the digital pumps are constant, if the working displacement of the digital motors is greater, then the rotational speed is slower, and the output torque is higher. Therefore, the output torque of digital motors can be controlled by changing the displacement ratio of digital pumps and motors.

After comparing the various hydraulic transformers and digital pumps/motors mentioned above, we propose a digital hydraulic transformer (DHT) [23], which is a new hydraulic secondary element that combines digital technology with traditional hydraulic elements. The design, modeling, simulation, and experimental verification of a traction hydraulic circuit are performed on the basis of this DHT. The flow/pressure control of the hydraulic system is realized through the accurate control of DHT binary numbers. This new type of DHT has the advantages of high reliability, simple command, no hysteresis, and good repeatability in control technology. Moreover, the hydraulic secondary element has simpler structure and larger pressure regulation range, and can realize continuous high pressure/large flow output.

## 2. Structural Composition and Pressure Transformation Characteristics of DHT

### 2.1. Structural Composition of DHT

The main body of DHT includes a flow dividing and collecting system composed of four gear–pump/motor units [24] and two sets of control valve group composed of four two-position, three-way solenoid directional valves [25]. The driving gear shafts of the four gear–pump/motor units are rigidly connected, and their displacements are sequentially arranged in accordance with the exponential law of two. Figure 1 presents the structure diagram of a flow dividing and collecting system composed of four gear–pump/motor units. Figure 2 shows a DHT that consists of four gear–pump/motor units and two control valve sets.

A control valve group is installed on the inlet and outlet sides of the four gear–pump/motor units, i.e., the inlet and outlet control valve groups. The inlet control valve group incorporates four two-position, three-way solenoid directional valves that control the input oil ports of the four gear–pump/motor units to switch between the oil port P and the oil port T. The working states of each two-position, three-way solenoid directional valve are represented by binary numbers “0” or “1”. When the state is “0”, the solenoid is de-energized and the input oil port of the corresponding gear–pump/motor unit is connected to the oil port T. When the state is “1”, the solenoid is energized and the input oil port of the corresponding gear–pump/motor unit is connected to the oil port P. The outlet control valve group also comprises four two-position, three-way solenoid directional valves, which control the output oil ports of the four gear–pump/motor units to switch between the oil port A and the oil port T. When the state of the two-position three-way solenoid directional valve is “0”, the solenoid is de-energized and the output oil port of the corresponding gear–pump/motor unit is connected to the oil port T. When the state is “1”, the solenoid is energized and the output oil port of the corresponding gear–pump/motor unit is connected to the oil port A.

### 2.2. Pressure Transformation Characteristics of DHT

The oil source ports of the four solenoid directional valves of the inlet control valve group are connected through the inlet valve block and summarized as external oil port P. The load oil ports of the four solenoid directional valves of the outlet control valve group are connected by the outlet valve block and summarized as external oil port A. The low-pressure oil ports of the eight solenoid directional valves on both sides of the DHT are connected by the inlet and outlet valve blocks, and they are summarized as external oil port T to balance the flow difference between oil ports P and A. The working principle diagram of this DHT is presented in Figure 3.

In accordance with the law of energy conservation, a variable displacement pump is used as the power element of DHT when analyzing its variable pressure principle in the ideal case. Meanwhile, the pressure and flow input to oil port P of DHT is used as the reference value. When the inlet and outlet control valve groups of DHT are all energized,
(1)$$p_{i}q_{i}=\frac{p_{o}}{15}q_{i}+\frac{2p_{o}}{15}q_{i}+\frac{4p_{o}}{15}q_{i}+\frac{8p_{o}}{15}q_{i},$$
where *p_i_* is the pressure of oil port P, *p_o_* is the pressure of oil port A, and *q_i_* is the flow of oil port P. As shown in Equation (1), the output pressure/flow of oil port A is equal to the input pressure/flow of oil port P at this moment, and the performance function of DHT is neither increasing nor decreasing pressure.

When all the control valves on the inlet side of DHT are powered, the outlet side solenoid directional valve “o1” is powered, whereas solenoid directional valves “o2”, “o3”, and “o4” are de-powered. Solenoid directional valves “o2”, “o3”, and “o4” output hydraulic oil back into the tank; output pressure is defaulted to 0; and pressure/flow output from solenoid directional valve “o1” enters the system. At this moment, gear–pump/motor units “m2”, “m3”, and “m4” are under motor working condition. They output torque to gear–pump/motor unit “m1”, and gear–pump/motor unit “m1” is under pump working condition. From the preceding description,
(2)$$p_{i}q_{i}=\frac{p_{o}}{15}q_{i}.$$

Equation (2) shows that the output pressure of oil port A is 15 times the input pressure of oil port P, the output flow of oil port A is 1/15 times the input flow of oil port P, and the DHT performance function is increasing pressure.

When solenoid directional valve “i1” on the inlet side of DHT is energized, solenoid directional valves “i2”, “i3”, and “i4” are de-energized, whereas solenoid directional valves on the outlet side are energized. Solenoid directional valves “i2”, “i3”, and “i4” draw oil from the tank into the corresponding gear–pump/motor units. At this moment, gear–pump/motor unit “m1” is under motor working condition and output torque to gear–pump/motor units “m2”, “m3”, and “m4”. Meanwhile, gear–pump/motor units “m2”, “m3”, and “m4” are under pump working condition. From the preceding description,
(3)$$p_{i}q_{i}=p_{o}q_{i}+2p_{o}q_{i}+4p_{o}q_{i}+8p_{o}q_{i}.$$

Equation (3) shows that the output pressure of oil port A is 1/15 times the input pressure of oil port P, the output flow of oil port A is 15 times the input flow of oil port P, and the DHT performance function is decreasing pressure.

The above scenario is part of the pressure transformation process of DHT, such that the power on/off states of solenoid directional valves “i1”, “i2”, “i3”, and “i4” on the inlet side correspond to a 4-bit binary number “*N_i_*”, while the power on/off states of solenoid directional valves “o1”, “o2”, “o3”, and “o4” on the outlet side correspond to a 4-bit binary number “*N_o_*”. The power on/off states of the control valve groups on the inlet and outlet sides are combined, and 225 types are derived, as indicated in Table 1.

From Table 1, the pressure/flow variation relationship between oil ports P and A is summarized as follows:(4)$$p_{o}=\frac{N_{i}}{N_{o}}p_{i},\;\;\;\;\;q_{o}=\frac{N_{o}}{N_{i}}q_{i},$$
where *q_o_* is the flow of oil port A. In accordance with Equation (4), the range of pressure transformation ratio of DHT can be theoretically reached [1/15, 15].

## 3. Modeling and Simulation of the Hydraulic System

### 3.1. Simulation and Analysis of DHT

Figure 4 shows the physical model of a digital speed-regulating traction hydraulic control system based on DHT. This model is constructed via joint simulation of AMESim and Simulink. To verify the rationality of the traction hydraulic system circuit design, we first adopted the control method of digital logic combination to encode and control eight solenoid directional valves to verify the feasibility of the DHT pressure transformation principle.

In the hydraulic system shown in Figure 4, solenoid on/off valve “5.3” is in the power-on state. Hydraulic oil flows into the tank through DHT, the variable throttle valve “10”, and the two-position, three-way solenoid directional valve “8.2”. The other solenoid directional valves are in the power-off state, as mentioned earlier, and the hydraulic system simulation model is based on the throttle valve as the simulated load. The relationship between flow and pressure of this variable throttle valve is shown in Equation (5),
(5)$$q=C_{d}A_{T}\sqrt{\frac{2}{\rho }\left(p_{1}-p_{2}\right)},$$
where *C_d_* is the orifice flow coefficient of the throttle valve, *A_T_* is the orifice area of the throttle valve, *ρ* is the density of oil, *p*_1_ is the inlet pressure of the throttle valve, and *p*_2_ is the outlet pressure of the throttle valve. In accordance with Equation (5), the input/output flow of the throttle valve is positively related to the input pressure of the throttle valve when the orifice flow coefficient, orifice area, oil density, and outlet pressure of the throttle valve are certain. The input pressure of the throttle valve is the simulated load pressure of the system. When its input/output flow changes, load pressure also changes. This condition can effectively simulate the change in output pressure when DHT is digitally controlled under variable load conditions.

#### 3.1.1. Simulation Research on DHT Pressure Transformation Ratio as a Constant Value

In accordance with the digital logic combinations of the power on/off states of the eight solenoid directional valves in Table 1, DHT has 13 pressure transformation ratios among the 59 control coding methods, indicating the existence of multiple logic combination states under the same pressure transformation ratio. Table 2 presents various binary digital logic combination relationships for a DHT pressure transformation ratio of 1.

In the holding pressure stage, the digital control matrix of the solenoid directional valves on the DHT inlet/outlet is set as
(6)$$N_{i}^{\prime }=\left[\begin{array}{c} \begin{array}{c} N_{i1}\\ N_{i3} \end{array}\\ \begin{array}{c} N_{i5}\\ \begin{array}{c} N_{i7}\\ \begin{array}{l} N_{i9}\\ N_{i11}\\ N_{i13}\\ N_{i15} \end{array} \end{array} \end{array} \end{array}\right]=\left[\begin{array}{cccc} 1 & 0 & 0 & 0\\ 1 & 1 & 0 & 0\\ 1 & 0 & 1 & 0\\ 1 & 1 & 1 & 0\\ 1 & 0 & 0 & 1\\ 1 & 1 & 0 & 1\\ 1 & 0 & 1 & 1\\ 1 & 1 & 1 & 1 \end{array}\right],\;\;\;\;\;N_{o}^{\prime }=\left[\begin{array}{c} \begin{array}{c} N_{o1}\\ N_{o3} \end{array}\\ \begin{array}{c} N_{o5}\\ \begin{array}{c} N_{o7}\\ \begin{array}{l} N_{o9}\\ N_{o11}\\ N_{o13}\\ N_{o15} \end{array} \end{array} \end{array} \end{array}\right]=\left[\begin{array}{cccc} 1 & 0 & 0 & 0\\ 1 & 1 & 0 & 0\\ 1 & 0 & 1 & 0\\ 1 & 1 & 1 & 0\\ 1 & 0 & 0 & 1\\ 1 & 1 & 0 & 1\\ 1 & 0 & 1 & 1\\ 1 & 1 & 1 & 1 \end{array}\right].$$

Figure 5 shows the characteristic curves of DHT pressure, flow, and gear–pump/motor units’ rotational speed in the eight preceding power on/off logic combination states.

Curves 1–8 in Figure 5a,b represent the regulation characteristic curves of pressure and flow with time at oil port A, curve 9 in Figure 5a illustrates the adjustment characteristic curve of oil port P pressure with time, and curve 9 in Figure 5b provides the modification characteristic curve of the flow of tank oil into oil port T with time. The binary digit of the control valve groups corresponding to curve 1 is *N_i_*_1_ = 1000, *N_o_*_1_ = 1000. The binary logic combination of the eight solenoid directional valves corresponding to curve 8 is *N_i_*_15_ = 1111, *N_o_*_15_ = 1111. In accordance with the control matrix (6), curves 1–8 are the increasing displacement control of DHT; that is, the binary digit logic combination is set as *N_i_*_1_ = 1000, *N_o_*_1_ = 1000 when DHT displacement is *V*_0_. Then, the logic combination is set as *N_i_*_15_ = 1111, *N_o_*_15_ = 1111 when DHT displacement is 15 *V*_0_.

As shown in Figure 5a, the output pressure of oil port A is 0 MPa when DHT is starting, while the input pressure of oil port P is 5 MPa. In accordance with Equation (7), the greater the displacement of DHT, the higher the starting torque *T*_1_.
(7)$$\left\{ \begin{array}{l} T_{1}=V\cdot \Delta p\\ \Delta p=p_{P}-p_{A} \end{array}\right.,$$
where *T*_1_ is the torque of gear–pump/motor units, *V* is the displacement of DHT, Δ*p* is the pressure difference between oil ports P and A, *p_P_* is the input pressure of oil port P, and *p_A_* is the output pressure of oil port A.

The rotational inertia of DHT is a fixed value. Thus, in accordance with Equation (8), the torque *T*_1_ of DHT is proportional to the angular acceleration *α*, indicating that the higher the torque *T*_1_ of DHT, the greater the angular acceleration *α* and slope *k* of DHT rotational speed. In accordance with Figure 5b and Equation (5), when the output pressure of oil port A reaches 5 MPa, all the output flow values of oil port A under the eight combination states also reach the amplitude values, and the amplitude values are equal. In accordance with Equation (9), when the output flow is a constant value, the larger the DHT displacement, the smaller the achieved stable rotational speed *n*, as shown in Figure 5c.
(8)$$\left\{ \begin{array}{l} T_{1}=J\cdot \alpha \\ \alpha =\frac{2\pi \cdot \Delta n}{\Delta t}\\ k=\frac{\Delta n}{\Delta t} \end{array}\right.,$$
(9)$$q_{o}=V\cdot n,$$
where *J* is the rotational inertia of DHT, *α* is the angular acceleration of DHT, and *n* is the rotational speed of DHT. When DHT outputs pressure/flow with a pressure transformation ratio of 1, a conclusion is drawn through the logic combination control of each solenoid directional valve that the larger the DHT displacement, the shorter the rise time of its output pressure/flow. Moreover, the input flow of oil port P is equal to the output flow of oil port A, and the flow of the tank into oil port T is zero.

#### 3.1.2. Simulation Research on DHT Increasing Pressure Stage

When the control valve group on the inlet side of DHT is set to be fully energized, the control matrix of the control valve group on the outlet side is
(10)$$N_{o}^{\prime \prime }=\left[\begin{array}{c} \begin{array}{c} N_{o15}\\ N_{o13} \end{array}\\ \begin{array}{c} N_{o11}\\ \begin{array}{c} N_{o9}\\ \begin{array}{l} N_{o11}\\ N_{o13}\\ N_{o15} \end{array} \end{array} \end{array} \end{array}\right]=\left[\begin{array}{cccc} 1 & 1 & 1 & 1\\ 1 & 0 & 1 & 1\\ 1 & 1 & 0 & 1\\ 1 & 0 & 0 & 1\\ 1 & 1 & 0 & 1\\ 1 & 0 & 1 & 1\\ 1 & 1 & 1 & 1 \end{array}\right].$$

In accordance with the above control matrix, the ratios of DHT output pressure to input pressure are obtained as
(11)$$\lambda_{1}=\frac{p_{o}}{p_{i}}\in \left[\frac{15}{15},\frac{15}{13},\frac{15}{11},\frac{15}{9},\frac{15}{11},\frac{15}{13},\frac{15}{15}\right].$$

As shown in Figure 6a, the input pressure of oil port P is a fixed value, while the output pressure of oil port A changes in a stepped state with the power on/off controls of the outlet control valve group, realizing the increasing pressure regulation function. However, when the outlet control valve group is switching the power on/off states, the output pressure of oil port A exhibits a negative shock in the increasing–increasing pressure stage and a positive shock in the increasing–decreasing pressure stage. The increasing–increasing pressure phase can be seen in Figure 6b. When the solenoid directional valves on the outlet side of DHT are switched at the moment of decreasing displacement, the change amount Δ*q_i_* of output flow at oil port A is
(12)$$\Delta q_{i}=\Delta V_{i}\cdot n_{i},$$
where Δ*V_i_* is the amount of change in DHT displacement each time the control valve group is switched, and *n_i_* is the rotational speed of DHT at the moment when the control valve group is switched.

The change amount of output flow at oil port A drops sharply by Δ*q_i_*, and in accordance with Equation (5), the output pressure of oil port A is positively correlated with output flow, resulting in a negative shock to the output pressure at oil port A.

In the increasing–decreasing pressure stage, the control valve group is an incremental displacement switch, and the flow at oil port A increases instantaneously by Δ*q_i_* at the instant of switching, resulting in a positive shock to output pressure. As shown in Figure 6c, the rotational speed of DHT initially rises in a stepped state and then falls in a stepped state. The rotational speed rising in a stepped state is the increasing–increasing pressure stage of DHT. The power on/off switching of the outlet control valve group results in a part of the gear–pump/motor units changing from the pump working condition to the motor working condition, indicating that a part of the hydraulic energy is transformed into mechanical energy, increasing the rotational speed of DHT. The rotational speed adjustment of DHT in the increasing–decreasing pressure stage is the opposite of the rotational speed regulation of DHT in the increasing–increasing pressure stage.

#### 3.1.3. Simulation Research on DHT Decreasing Pressure Stage

Setting the control valve group on the outlet side of DHT to be fully energized in the decreasing pressure stage, the control matrix of the control valve group on the inlet side of DHT is
(13)$$N_{i}^{\prime \prime }=\left[\begin{array}{c} \begin{array}{c} N_{i15}\\ N_{i13} \end{array}\\ \begin{array}{c} N_{i11}\\ \begin{array}{c} N_{i9}\\ \begin{array}{l} N_{i11}\\ N_{i13}\\ N_{i15} \end{array} \end{array} \end{array} \end{array}\right]=\left[\begin{array}{cccc} 1 & 1 & 1 & 1\\ 1 & 0 & 1 & 1\\ 1 & 1 & 0 & 1\\ 1 & 0 & 0 & 1\\ 1 & 1 & 0 & 1\\ 1 & 0 & 1 & 1\\ 1 & 1 & 1 & 1 \end{array}\right].$$

In accordance with the above control matrix (13), the ratios of DHT output pressure to input pressure are obtained as
(14)$$\lambda_{2}=\frac{p_{o}}{p_{i}}\in \left[\frac{15}{15},\frac{13}{15},\frac{11}{15},\frac{9}{15},\frac{11}{15},\frac{13}{15},\frac{15}{15}\right].$$

In accordance with the power on/off controls of the matrix (13), the pressure, flow, and rotational speed of DHT in the hydraulic system are depicted in Figure 7. As shown in Figure 7a,b, the input pressure of oil port P is constant and the output pressure of oil port A is changed in a stepped state. DHT realizes the function of decreasing pressure regulation. When DHT is regulated by decreasing pressure, the difference value between the output flow *q_o_* at oil port A and the input flow *q_i_* at oil port P is the flow *q_t_* of the tank into oil port T, achieving the objective of decreasing pressure and increasing flow.

When DHT is regulated by decreasing pressure, the output pressure of oil port A does not exhibit the shocking phenomenon. The primary reason for this result is that the state of the DHT inlet side control valve group gain/loss of power is changed in the process of variable pressure. Meanwhile, the control valve group on the outlet side is always energized. At the moment when the control valve set on the inlet side is switched, the rotational speed of DHT does not change, namely, the output flow of DHT oil port A does not change, and thus, the shock phenomenon does not occur. As shown in Figure 7c, the rotational speed of DHT initially steps down and then steps up. The rotational speed in the stepped down state is in the decreasing–decreasing pressure phase of DHT. The primary reason for this finding is that when the control valve group on the DHT inlet side is combined with the power on/off states, a part of the gear–pump/motor units is changed from the motor working condition to the pump working condition. Then, mechanical energy is partially converted into hydraulic energy, reducing the rotational speed of gear–pump/motor units. The rotational speed adjustment of DHT in the decreasing–increasing pressure phase is opposite to the rotational speed regulation of DHT in the decreasing–decreasing pressure stage.

### 3.2. Simulation and Analysis of the Hydraulic System Based on DHT

When the hydraulic cylinder piston rod rises, the oil source outputs high-pressure and low-flow hydraulic oil through DHT to low-pressure and high-flow, and the DHT positive rotation, which behaves as a decreasing pressure function. At this moment, solenoid on/off valves 5.1 and 5.3 and solenoid directional valve 8.1 are energized. By contrast, solenoid on/off valve 5.2 and solenoid directional valves 8.3 and 8.4 are de-energized. That is, the hydraulic traction system starts to provide oil pressure from the bladder accumulator. When oil pressure of the bladder accumulator decreases to a point when the hydraulic cylinder piston rod’s operating speed can no longer be maintained by adjusting DHT, the hydraulic system oil pressure will be provided by the constant-pressure variable displacement pump, while the bladder accumulator will no longer provide oil pressure. When the hydraulic cylinder piston rod descends, solenoid on/off valve 5.2 and solenoid directional valves 8.3 and 8.4 are energized. By contrast, solenoid on/off valves 5.1 and 5.3 and solenoid directional valve 8.1 are de-energized. The hydraulic traction system oil pressure is provided by the constant-pressure variable displacement pump. The hydraulic cylinder rodless chamber oil pressure is regulated by DHT at this moment, and DHT reverses rotation. When the variable pressure ratio of DHT increases, the pressure of the hydraulic oil entering DHT decreases. That is, the back pressure of the hydraulic cylinder rodless chamber decreases. The hydraulic cylinder piston rod accelerates downward at this moment, and part of the hydraulic oil in the rodless chamber of the hydraulic cylinder is returned to the tank through oil port T of DHT. Meanwhile, another part of the hydraulic oil enters the bladder accumulator through port P of DHT to realize energy recovery. The major parameters of the physical model of the hydraulic traction system are provided in Table 3.

The hydraulic cylinder piston rod rise and fall displacement curves are shown in Figure 8a,b. The maximum displacement of the hydraulic cylinder piston rod is 3.6 m when it rises and falls. In the ascending stage, the hydraulic cylinder piston rod extends, and the time delay of the following curve is the longest when displacement is 0.5 m, time delay *t*_1_ = 2.9 s, but the displacement of the hydraulic cylinder piston rod gradually increases with time, and time delay gradually decreases. In the descending stage, the hydraulic cylinder piston rod retracts, and the time delay of the following curve is the longest when the displacement is 0.5 m, and the time delay is *t*_2_ = 2.5 s. The displacement of the hydraulic cylinder piston rod gradually increases with time, and time delay gradually decreases similar to that in the ascending stage. Figure 8c shows a graph of the velocity curve of the hydraulic cylinder piston rod. Curve 1 is the velocity profile when the hydraulic cylinder piston rod is rising, and curve 2 is the velocity profile when it is falling. The maximum running speed of the hydraulic cylinder piston rod when rising is *v*_1_, and speed *v*_1_ = 0.22 m/s. The maximum running speed of the hydraulic cylinder piston rod when falling is *v*_2_, and speed *v*_2_ = −0.23 m/s. The difference between the rising and falling times of the hydraulic cylinder piston rod is *t*_3_, and the difference *t*_3_ = 1 s. A graph of the hydraulic system oil source output/input pressure curve is presented in Figure 8d. Curve 1 is the output pressure profile of the bladder accumulator. When the hydraulic cylinder piston rod starts to rise, pressure is initially output by the bladder accumulator, and pressure is 20 MPa. When the output pressure of the bladder accumulator drops to 9 MPa, the hydraulic system oil pressure is then provided by the constant-pressure variable displacement pump. As shown in curve 3, the output pressure value is 9 MPa. Curve 2 shows the storage pressure change curve in the bladder accumulator when the hydraulic cylinder piston rod is descending.

The simulation of the hydraulic traction system model based on DHT shows that the speed regulation of the hydraulic cylinder piston rod in the hydraulic traction system can be achieved through binary digital control of the power on/off states of the eight solenoid directional valves in DHT. In addition, during the piston rod descent of the hydraulic cylinder, DHT converts pressure oil output from the rodless chamber of the hydraulic cylinder from a low-pressure and high-flow state into a high-pressure and low-flow state. The pressure oil enters the bladder accumulator, completing the energy recovery of the hydraulic system.

## 4. Experimental Research on the Hydraulic System

Figure 9 presents the experimental simulation platform of the hydraulic traction system based on DHT. On the one hand, this experiment reduces the output pressure of the constant-pressure variable pump step by step, simulating the output pressure of the bladder accumulator when the piston rod of the hydraulic cylinder rises. Moreover, the speed control of the hydraulic cylinder piston rod in the hydraulic system is experimentally studied through the binary digital control of the output flow/pressure of DHT. On the other hand, the output pressure of the constant-pressure variable pump is a fixed value. By adjusting the set value of the relief valve, the back pressure of the hydraulic cylinder’s oil return chamber is changed to simulate the changing load of the hydraulic cylinder. Moreover, the speed control of the hydraulic cylinder piston rod in the hydraulic system is experimentally studied through the binary digital control of the output flow/pressure of DHT.

Figure 10a–c show the speed variation curves of the hydraulic cylinder piston rod when the oil pressure supplied by the constant-pressure variable pump is 7, 6, and 5 MPa, respectively. The setting value of relief valve 5.2 is 3.5 MPa; that is, the load on the piston rod of the hydraulic cylinder is a fixed value. As shown in Figure 10a, the maximum movement speed value of the hydraulic cylinder piston rod when it is extending is *v_maxa_*_1_, *v_maxa_*_1_ = 0.020 m/s, while the maximum movement speed value when it is retracting is *v_maxa_*_2_, *v_maxa_*_2_ = −0.022 m/s. At this moment, the binary control numbers of the eight solenoid directional valves of DHT are *N_i_*_1_ = 1101, *N_o_*_1_ = 1111. As the output pressure of the constant-pressure variable pump decreases, the maximum movement speed of the hydraulic cylinder piston rod when it is extended remains basically unchanged compared with that in Figure 10a, as shown in Figure 10b,c, by controlling the digital combination of the eight solenoid directional valves, i.e., *N_i_*_2_ = 1011, *N_o_*_2_ = 1111, *N_i_*_3_ = 1111, *N_o_*_3_ = 1111. When it retracts, the maximum velocities in Figure 10b,c are slightly higher compared with Figure 10a. However, the difference value of that velocity change is relatively small.

The output pressure value of constant-pressure variable displacement pump 4 is 5 MPa, and the load pressure values are 3.5, 3.0, and 2.5 MPa. As shown in Figure 11, the power on/off binary numbers of the eight solenoid directional valves of DHT in the hydraulic traction system are *N_i_*_3_ = 1111, *N_o_*_3_ = 1111, *N_i_*_4_ = 0011, *N_o_*_4 =_ 1111, *N_i_*_5_ = 1001, *N_o_*_5_ = 1111. As load pressure gradually decreases, the maximum speed of the hydraulic cylinder piston rod extension/retraction also gradually decreases, but the difference in maximum speed is relatively small. In addition, with the gradual decrease in load pressure value, the fluctuation of the maximum movement speed change curve becomes larger but can still be disregarded.

## 5. Conclusions

A new configuration for a hydraulic transformer, namely, a digital hydraulic transformer, is proposed for the CPR secondary regulation system. DHT is formed by rigidly connecting the output shafts of four gear–pump/motor units with a displacement ratio of 1:2:4:8, and eight solenoid directional valves are installed on the inlet and outlet of gear–pump/motor units. Through the binary digital control of the DHT inlet and outlet control valve groups, changing the flow direction and size of oil port T is possible. In turn, such change alters the output flow of oil port A and achieves the objective of changing the output pressure of oil port A. A theoretical study of DHT is performed, the variable pressure principle of DHT is analyzed, and the binary digital combination states of the DHT inlet and outlet control valve groups are summarized. A conclusion is drawn that this DHT, which is composed of four gear–pump/motor units, has 225 control coding methods, and ideally, an output pressure range of 1/15 to 15 times relative to the input pressure can be achieved.

A dynamic simulation model of the hydraulic traction system based on DHT is established, and the pressure/flow of DHT and the hydraulic traction system is simulated. A conclusion is drawn that the speed and displacement of the hydraulic cylinder piston rod can be controlled through the binary digital control of the eight solenoid directional valves, verifying the rationality of the design circuit of the hydraulic traction system. Finally, the speed control of the hydraulic cylinder piston rod of the hydraulic traction system based on DHT is experimentally verified by relying on an experimental simulation platform. The results show that the designed digital hydraulic transformer can change the output flow of the hydraulic traction system through digital control.

## Figures and Tables

**Figure 1 sensors-22-03624-f001:**
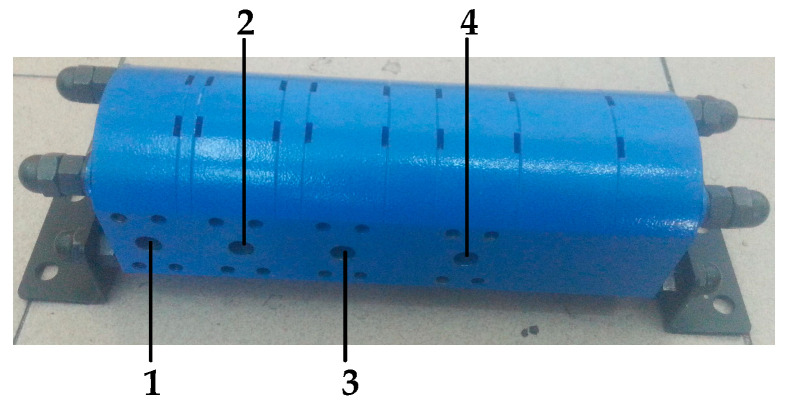
Structure diagram of the flow dividing and collecting system of four gear–pump/motor units. (1) Gear–pump/motor with a displacement of *V*_0_; (2) Gear–pump/motor with a displacement of 2 *V*_0_; (3) Gear–pump/motor with a displacement of 4 *V*_0_; (4) Gear–pump/motor with a displacement of 8 *V*_0_.

**Figure 2 sensors-22-03624-f002:**
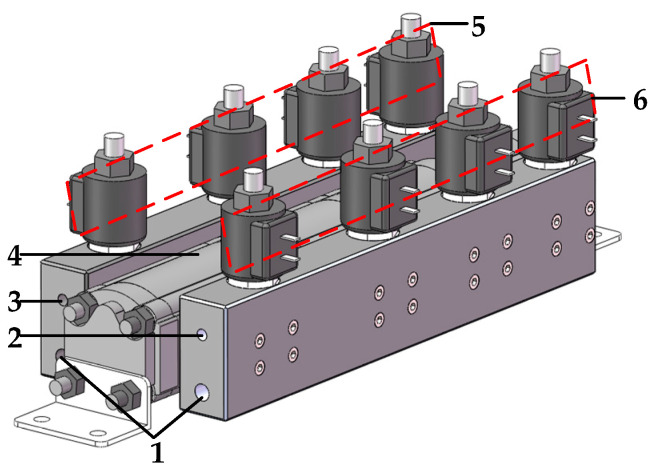
Structure diagram of DHT. (1) Low-pressure port T for connecting the tank; (2) Oil port P for connecting the oil source; (3) Oil port A for connecting the load; (4) Flow dividing and collecting system; (5) Outlet control valve group; (6) Inlet control valve group.

**Figure 3 sensors-22-03624-f003:**
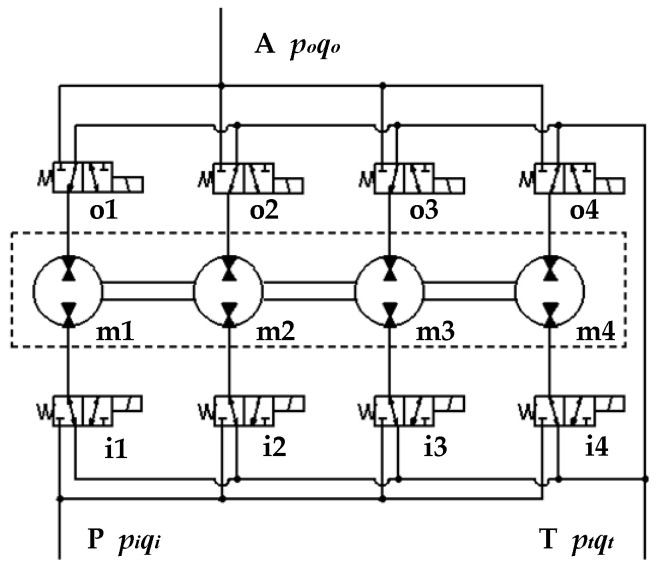
Working principle diagram of DHT.

**Figure 4 sensors-22-03624-f004:**
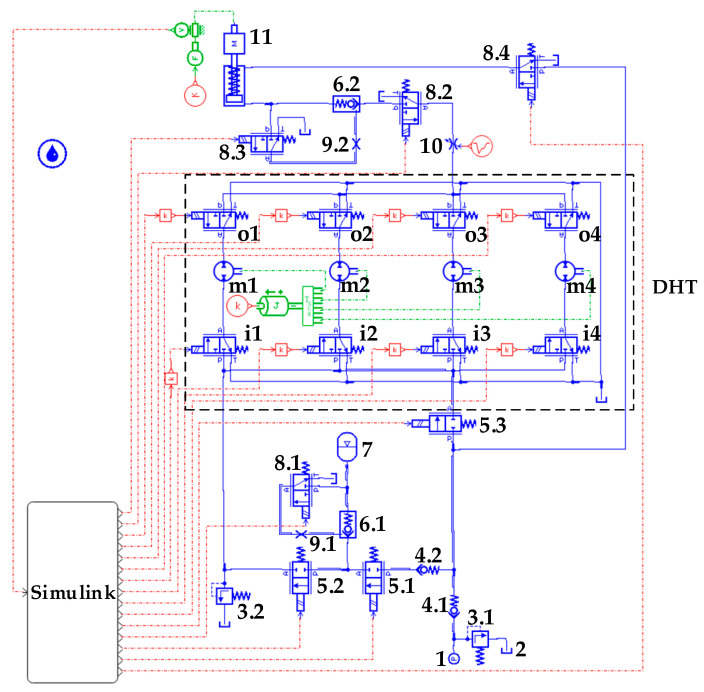
Physical simulation model of the hydraulic circuit of the traction system. (1) Pressure source. (2) Tank. (3) Relief valve. (4) Check valve. (5) Solenoid on/off valve. (6) Hydraulic controlled check valve. (7) Bladder accumulator. (8) Two-position, three-way solenoid directional valve. (9) Throttle valve. (10) Variable throttle valve. (11) Hydraulic cylinder. In addition, when using the oil pressure to DHT oil port P as the reference value and regulating its pressure, the pressure output from DHT can be divided into three stages: increasing, holding, and decreasing pressure. In the increasing pressure phase, when the output pressure of DHT exhibits upward, constant, and downward trends, the processes of DHT pressure change are called increasing–increasing pressure, increasing–holding pressure, and increasing–decreasing pressure regulation, respectively. In the holding pressure stage, the variable pressure ratio of DHT is 1 and the input pressure of DHT is equal to the output pressure under ideal conditions. In the decreasing pressure stage, when the output pressure of DHT exhibits rising, unchanging, and falling trends, the process of DHT pressure change is called decreasing–increasing pressure, decreasing–holding pressure, and decreasing–decreasing pressure regulation, respectively.

**Figure 5 sensors-22-03624-f005:**
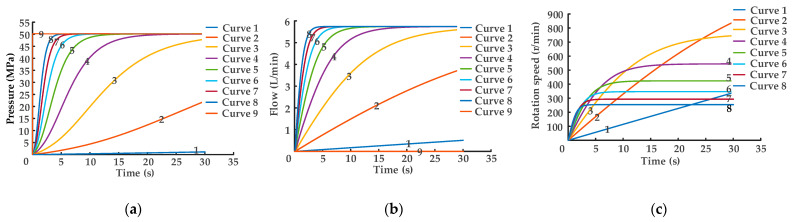
Regulation characteristic curves of DHT in the holding pressure stage: (**a**) pressure regulation; (**b**) flow regulation; (**c**) gear–pump/motor units’ rotational speed characteristic curves.

**Figure 6 sensors-22-03624-f006:**
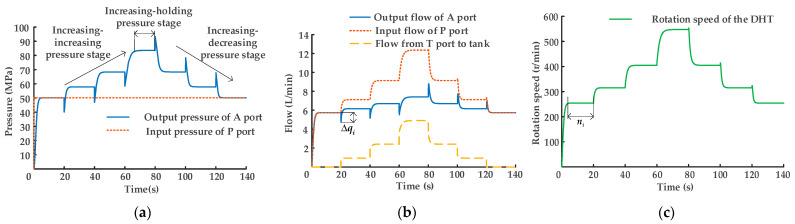
Regulation characteristic curves of DHT in increasing pressure stage: (**a**) pressure regulation; (**b**) flow regulation; (**c**) gear–pump/motor unit rotational speed characteristic curves.

**Figure 7 sensors-22-03624-f007:**
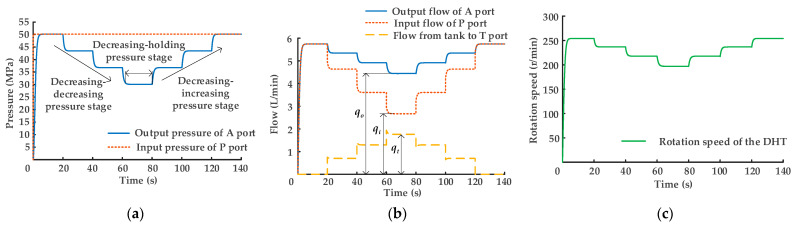
Regulation characteristic curves of DHT in the decreasing pressure stage: (**a**) pressure regulation; (**b**) flow regulation; (**c**) gear–pump/motor unit rotational speed characteristic curves.

**Figure 8 sensors-22-03624-f008:**
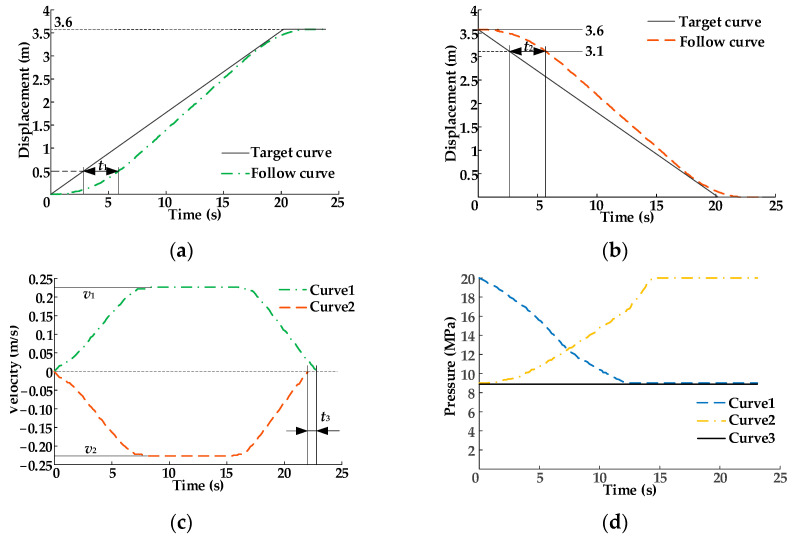
Regulation characteristic curves of the hydraulic traction system: (**a**) rising displacement curves of hydraulic cylinder piston rod; (**b**) descending displacement curves of the hydraulic cylinder piston rod; (**c**) velocity curves of the hydraulic cylinder piston rod; (**d**) oil source output/input pressure curves of the hydraulic traction system.

**Figure 9 sensors-22-03624-f009:**
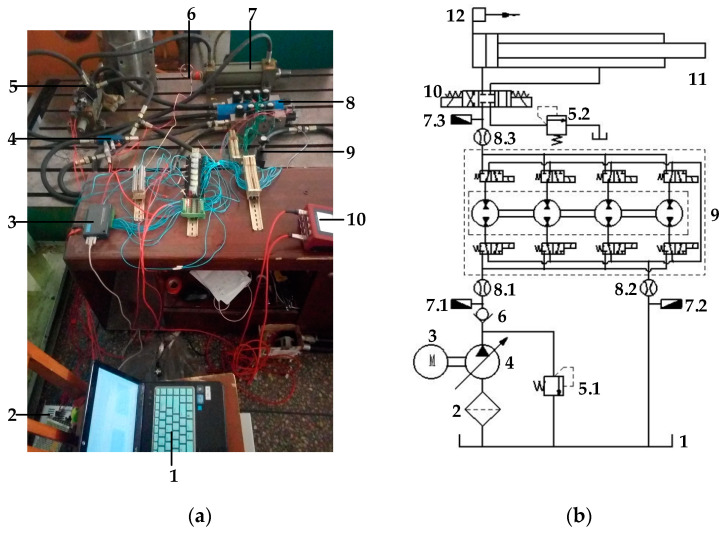
Experimental simulation platform of the hydraulic traction system. (**a**) Physical diagram of the experiment platform. (1) Measurement and control system. (2) Direct current power. (3) Data acquisition card. (4) Flow/pressure sensor. (5) Three-position, four-way solenoid directional valve. (6) Displacement sensor. (7) Hydraulic cylinder. (8) DHT. (9) Relief valve. (10) Flow/pressure gauge; (**b**) Hydraulic schematic of the experimental platform. (1) Tank. (2) Filter. (3) AC motor. (4) Constant-pressure variable displacement pump. (5) Relief valve. (6) Check valve. (7) Pressure sensor. (8) Flow sensor. (9) DHT. (10) Three-position, four-way solenoid directional valve. (11) Hydraulic cylinder. (12) Displacement sensor.

**Figure 10 sensors-22-03624-f010:**
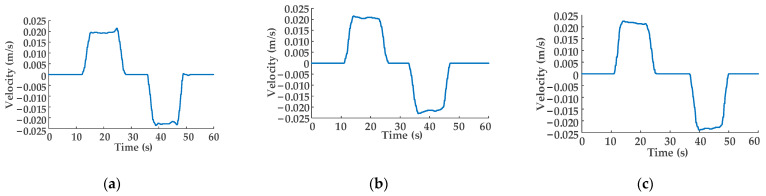
Speed variation curves of the hydraulic cylinder piston rod: (**a**) output pressure of the constant-pressure variable pump is 7 MPa; (**b**) output pressure of the constant-pressure variable pump is 6 MPa; (**c**) output pressure of the constant-pressure variable pump is 5 MPa.

**Figure 11 sensors-22-03624-f011:**
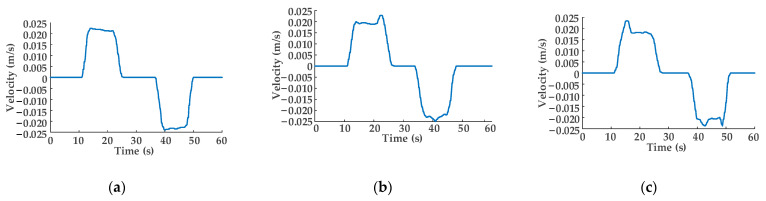
Speed variation curves of the hydraulic cylinder piston rod: the pressure setting values of relief valve 5.2 are (**a**) 3.5 MPa; (**b**) 3.0 MPa; (**c**) 2.5 MPa.

**Table 1 sensors-22-03624-t001:** Combination table of DHT control valve group states.

Inlet/Outlet	Port A Pressure	Port A Flow	Inlet/Outlet	Port A Pressure	Port A Flow	⋅⋅⋅⋅⋅⋅
1000/1000	1/1 *p_i_*	1/1 *q_i_*	0100/1000	2/1 *p_i_*	1/2 *q_i_*	⋅⋅⋅⋅⋅⋅
1000/0100	1/2 *p_i_*	2/1 *q_i_*	0100/0100	2/2 *p_i_*	2/2 *q_i_*	⋅⋅⋅⋅⋅⋅
1000/1100	1/3 *p_i_*	3/1 *q_i_*	0100/1100	2/3 *p_i_*	3/2 *q_i_*	⋅⋅⋅⋅⋅⋅
1000/0010	1/4 *p_i_*	4/1 *q_i_*	0100/0010	2/4 *p_i_*	4/2 q_i_	⋅⋅⋅⋅⋅⋅
⋅⋅⋅⋅⋅⋅	⋅⋅⋅⋅⋅⋅	⋅⋅⋅⋅⋅⋅	⋅⋅⋅⋅⋅⋅	⋅⋅⋅⋅⋅⋅	⋅⋅⋅⋅⋅⋅	⋅⋅⋅⋅⋅⋅
1000/1111	1/15 *p_i_*	15/1 *q_i_*	0100/1111	2/15 *p_i_*	15/2 *q_i_*	⋅⋅⋅⋅⋅⋅
⋅⋅⋅⋅⋅⋅	Inlet/Outlet	Port A Pressure	Port A Flow	Inlet/Outlet	Port A Pressure	Port A Flow
⋅⋅⋅⋅⋅⋅	0111/1000	14/1 *p_i_*	1/14 *q_i_*	1111/1000	15/1 *p_i_*	1/15 *q_i_*
⋅⋅⋅⋅⋅⋅	0111/0100	14/2 *p_i_*	2/14 *q_i_*	1111/0100	15/2 *p_i_*	2/15 *q_i_*
⋅⋅⋅⋅⋅⋅	0111/1100	14/3 *p_i_*	3/14 *q_i_*	1111/1100	15/3 *p_i_*	3/15 q_i_
⋅⋅⋅⋅⋅⋅	0111/0010	14/4 *p_i_*	4/14 *q_i_*	1111/0010	15/4 *p_i_*	4/15 *q_i_*
⋅⋅⋅⋅⋅⋅	⋅⋅⋅⋅⋅⋅	⋅⋅⋅⋅⋅⋅	⋅⋅⋅⋅⋅⋅	⋅⋅⋅⋅⋅⋅	⋅⋅⋅⋅⋅⋅	⋅⋅⋅⋅⋅⋅
⋅⋅⋅⋅⋅⋅	0111/1111	14/15 *p_i_*	15/14 *q_i_*	1111/1111	15/15 *p_i_*	15/15 *q_i_*

**Table 2 sensors-22-03624-t002:** Fifteen logic combinations with a pressure transformation ratio of 1.

Inlet/Outlet	Port A Pressure	Port A Flow
1000/1000	1/1 *p_i_*	1/1 *q_i_*
1100/0100	2/2 *p_i_*	2/2 *q_i_*
1100/1100	3/3 *p_i_*	3/3 *q_i_*
0010/0010	4/4 *p_i_*	4/4 *q_i_*
⋅⋅⋅⋅⋅⋅	⋅⋅⋅⋅⋅⋅	⋅⋅⋅⋅⋅⋅
0111/0111	14/14 *p_i_*	14/14 *q_i_*
1111/1111	15/15 *p_i_*	15/15 *q_i_*

**Table 3 sensors-22-03624-t003:** Major parameters of the hydraulic system.

System Parameter	Values
Piston diameter *D* (mm)	110
Piston rod diameter *d* (mm)	70
Piston stroke *S* (mm)	4000
Load mass *m* (kg)	2000
Accumulator volume *V* (L)	20
Relief valve 3.1 set pressure *p*_1_ (MPa)	9
Relief valve 3.2 set pressure *p*_2_ (MPa)	20
Ratio of four gear–pump/motor displacements *V*_1_:*V*_2_:*V*_3_:*V*_4_	1:2:4:8

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
