# Peer review of "Analysis of and Experimental Research on a Hydraulic Traction System Based on a Digital Hydraulic Transformer"

_sensors, 2022, doi:10.3390/s22103624_

Round 1

Reviewer 1 Report

OVERVIEW

The authors deal with a new type of digital hydraulic transformer. The authors propose a new energy-saving hydraulic system for the common pressure rail secondary regulation system based on a digital hydraulic transformer. The authors performed a theoretical study of the digital hydraulic transformer, analysed the variable pressure principle of the digital hydraulic transformer, and summarized the binary digital combination states of the digital hydraulic transformer inlet and outlet control valve groups. The authors established a dynamic simulation model of the hydraulic elevator system based on the digital hydraulic transformer. The authors concluded that the proposed digital hydraulic transformer can change the output pressure/flow of a hydraulic system through digital control.

POSITIVE ASPECTS

1. Based on a literature review, the authors made an overview of the issues related to hydraulic transformers, their advantages, and disadvantages.
2. The authors made a detailed description of the principle of operation of the digital hydraulic transformer.
3. The authors established a dynamic simulation model of the hydraulic elevator system based on a digital hydraulic transformer.
4. The authors summarized the binary digital combination states of the digital hydraulic transformer inlet and outlet control valve groups.
5. The authors established a dynamic simulation model of the hydraulic elevator system based on the digital hydraulic transformer.
6. The authors simulated the pressure/flow of the digital hydraulic transformer and the hydraulic elevator system.

ISSUES

The presented work is useful but has some issues that need to be removed. I have a few comments that can be used to improve the article.

Minor issues
1. Sections 2.1 and 2.2 have the same title. The section names need to be corrected according to their main idea.
2. In the description in Figure 2, the abbreviations T, P, and A are not indicated but are shown in Figure 3. Correct.
3. Signs of physical quantities need to write in italics according to ISO 31-4 (ISO 80000-5: 2007). Corrections need to be done throughout the article.
4. Equation 5 is missing a coherent introductory text. Complete the missing text before Equation 5.
5. The yellow line in the graph in Figure 7b should be dashed as indicated in the legend. Correct.

Major issues
1. The slope k on page 8 is not shown anywhere, neither in the equation nor in the graph. Fill in the missing information for the slope k in the equation or graph.

CONCLUSION

I find this review helpful. The authors of the present article have to correct the issues.

Reviewer 2 Report

sensors-1676758 -Peer-Review-V1

Analysis of and Experimental Research on a Hydraulic Traction System Based on a Digital Hydraulic Transformer

By Weijian Li, Zhuxin Zhang,*, Tao Liu, Hengyi Cao, Tao Ni, and Yafei Wang

This paper proposed a design of a new type of digital hydraulic transformer which combines digital technology with traditional hydraulic elements. The overall structure of the paper is fine. Both simulation tests and experimental tests are conducted to verify the effectiveness of the proposed design structure. However, the main advantage of the proposed DHT is not obvious. Besides, I have the following comments for your consideration.

  1. The main motivation of the proposed design should be highlighted. We can see that the authors have succeeded in changing the output pressure/flow of a hydraulic system through digital control. However, why do we need to make such changes? What is the advantage of utilizing the digital control should be emphasized.

  1. Following the comments above, there should be a comparison test to verify the merits of the digital control DHT.

  1. Introduction part is too long. Much emphasis is put on the introduction of traditional hydraulic transformer. However, since the research focus is DHT, there should be more description on DHT. Or at least, the authors should describe the disadvantages of traditional hydraulic transformers compared to DHT.

  1. Too many long sentences in the paper. Some of them with grammar errors. For example, the last sentence in Abstract is 7-line long. Please do not use such long sentences anymore! Only make readers painful.

  1. Page 5, Section 3.1. Please put more description on the simulation conditions. For example, what software is used to perform simulation tests? And how to setup the simulation parameters?

Round 2

Reviewer 2 Report

I have no other comments.